# The prognosis of different distant metastases pattern in malignant tumors of the adrenal glands: A population-based retrospective study

Jia Miao[1]☯, Haibin Wei[2]*, Jianxin Cui[2]☯, Qi Zhang[2], Feng Liu[2], Zujie Mao[2], Dahong Zhang📍[2]*

1 Department of Urology, Taizhou First People's Hospital, Taizhou, Zhejiang, China, 2 Department of Urology, Zhejiang Provincial People's Hospital, People's Hospital of Hangzhou Medical College, Hangzhou, Zhejiang, China

☯ These authors contributed equally to this work.
* whb-sysu@163.com (HW); zhangdahong88@yeah.net (DZ)

**Data Availability Statement:** The datasets analyzed during the current study are available in the SEER database, https://seer.cancer.gov/. The SEER database is a publicly available. Application

## Abstract

### Introduction

The present existing data on the association of metastatic sites and prognosis of patients with metastatic adrenal malignancy are limited. This study aims to investigate the impact of different distant metastases pattern on the survival of patients with adrenal malignancy.

### Methods

A dataset from the National Cancer Institute's Surveillance, Epidemiology, and End Results (SEER) 18 Registries (2000–2017) was selected for a retrospective metastatic adrenal malignancy cohort study. There was information on distribution of metastatic lesions in bone, brain, liver, and lung in the SEER database. Kaplan-Meier analysis and nomogram analyses were applied to compare the survival distribution of cases. Univariate and multivariate cox regression models were used to analyze survival outcomes.

### Results

From the SEER database, a total of 980 patients with primary metastatic adrenal malignancy from 2010 to 2017 were enrolled in this cohort study. Based on the initial metastatic sites, 42.3%, 38.4%, 30.5%, and 4.9% of patients were found bone, liver, lung, and brain metastasis, respectively. Patients who had a single site of distant metastases accounted for 52.6% (515/980) and had a better overall survival (OS) and cancer-specific survival (CSS) (both P < 0.001). In contrast with the tumor arising from the cortex, the tumor from the medulla showed better survival outcomes in both OS and CSS (P < 0.001).

methods can be found at https://seer.cancer.gov/data/access.html. A signed SEER Data-Use Agreement form is required to access the SEER data. Request forms may be accessed at https://seer.cancer.gov/seertrack/data/request. The SEER Program will process the request within 2 business days. We did not have any special access privileges. Any interested researchers can replicate our study by directly obtaining the data from the SEER database and following the protocol in our Methods section after gaining access.

**Funding:** This study was supported by the Medical Scientific Research Foundation of Zhejiang Province (2018KY019 and 2021KY449).

**Competing interests:** The authors have declared that no competing interests exist.

## Conclusion

Different histological types possess various metastatic features and prognostic values. Understanding these differences may contribute to designing targeted pre-treatment assessment of primary metastatic adrenal malignancy and creating a personalized curative intervention.

## Introduction

Adrenal tumor is a rare malignancy that is infrequently encountered by oncologists, primary care physicians, and urologists. Although adrenal incidentalomas have been increasingly identified by more abdominal cross-sectional imaging studies, only 8% of adrenal incidentalomas are malignant [1, 2]. Although adrenal cortical carcinoma (ACC) is the most common histological type of adrenal malignancy, its annual incidence is 0.7–2 cases every year and its worldwide prevalence is 4 to 12 cases per 100,000 individuals every year [1, 3]. En bloc resection with negative margins (R0 resection) may be a better choice for patients with respectable adrenal tumor, but R0 resection is not possible in many patients, especially patients with tumor metastasis [4]. Further, 50% to 75% of adrenal masses may represent metastatic disease in patients with known cancer [5, 6]. Visceral metastases of adrenal malignancy might be associated with worse outcomes. Diagnosis and treatment do not equate with equivalent benefit for metastatic adrenal malignancy, and further understanding of outcome of adrenal malignancy, especially metastatic adrenal malignancy, might help make reasonable medical decision and save the unnecessary expend on the advanced tumor. Several studies have focused on evaluating the prognosis of metastases in primary metastatic adrenal malignancy [7–9]. However, to date little attention has been focused on the prognostic significance of the distant metastatic pattern of adrenal malignancy at the first diagnosis.

Currently, the relative research on primary metastatic adrenal malignancy is very limited, and little information available is derived from these case series and case reports [10, 11]. To our knowledge, there are no prior studies with a large sample size that investigated the potential prognostic value of site-specific metastases in adrenal malignancies. Since knowledge of prognosis of the different distant metastatic pattern has an important role in pre-treatment evaluation, our study aimed to describe the distant metastatic site, frequency of occurrence, and pattern of these metastases based on a large population by using an established national cancer registry, the Surveillance, Epidemiology, and End Results (SEER) database. The goal of this study was to provide a better understanding of the prognosis of the different distant metastatic pattern in adrenal malignancy patients and help in making a suitable clinical decision at the first diagnosis.

## Materials and methods

### Data source and patient selection

The data of this study were obtained from the National Cancer Institute's SEER program dataset, a population-based cancer registry covering approximately 34.6% of the population in the United States. SEER*Stat software (SEER*Stat 8.3.6) was utilized to select eligible patients and define variables. The dataset from SEER Research Data, 18 Registries, Nov 2019 Sub (2000–2017) was selected for a retrospective primary metastatic adrenal malignancy cohort study. In this study, patients with primary metastatic adrenal malignancy were selected using inclusion

and exclusion rules from 2010 to 2017. The inclusion criteria were as follows: (1) information on distant metastases in the enrolled patients should be confirmed, including bone, brain, liver, and lung; (2) the diagnosis was confirmed clinically or pathologically with confirmed age. Patients were excluded if the age and the survival months were unknown, or the oncological behaviors were not malignant.

Based on the ICD-O-3, we divided the histological type into the following three subgroups: ACC, Neuroblastoma (NE), and others. Residential areas were also divided into the following three subgroups: metropolis, nonmetropolis, and unknown.

## Statistical analysis

The distribution of demographic factors and tumor characteristics were summarized with descriptive statistics. Categorical variables were compared using chi-square tests, and continuous variables were analyzed by the Mann Whitney test for non-normal distribution. Overall survival (OS) was calculated from the date of diagnosis to the date of death without limitation of any cause of death, or the date of the last follow-up. Cancer-specific survival (CSS) was defined as the interval from the date of diagnosis to death due to adrenal malignancy other than extra causes. Survival was analyzed by Kaplan-Meier methods, and the log-rank test was performed to evaluate the discrepancies between OS and CSS. The Cox proportional hazards regression model with the hazard ratio (HR) and the associated 95% confidence interval (CI) were utilized to evaluate the comparative risks of mortality. All statistical significance was 2-sided, $P < 0.05$ was considered statistically significant. All the above methods were performed in Statistics software version SPSS 25.0 (IBM, NY, US).

## Nomogram construction

A nomogram was built considered the factors of multivariate analysis and data availability. The maximum score for variable was 100 each. The performance of the nomogram was measured by concordance index (C-index) [12, 13]. Nomogram was performed in R version 4.0.0 software. (The R Foundation for Statistical Computing, Vienna, Austria. http://www.r-project.org).

## Results

### Patient demographics and clinical characteristics

A total of 2053 eligible patients with primary adrenal malignancy between 2010 and 2017 were identified from SEER database. Among these patients, the metastatic adrenal malignancy accounted for 47.74% (980/2053) at the time of diagnosis, including 493 men (50.3%) and 487 women (49.7%). The metastatic information on bone, brain, liver, and lung metastasis was collected in the SEER database.

The demographic characteristics of these patients are presented. (Table 1). The median age at diagnosis of the whole group was 21 years. Approximately 77.9% of patients' ethnicity was white. The distribution of the number of patients annually (from 2010 to 2017) showed no significant difference. Primary adrenal malignancy on the left side was more common than that on the right side, especially in NE. In addition to the grade of unknown, the most common grade was poor differentiated (30.9%). Approximately more than half of the patients (71.8%) were from middle-income families. Most of the patients (90.1%) lived in metropolis. 48.3% of patients had tumor size ≤ 100mm, but 19.1% had unknown tumor size. According to the histological type, the clinicopathological information of patients is also shown (Table 1). Because the population of ACC and NE was large, the group of histology was divided into three types.

**Table 1. Demographical characteristics of patients.**

| Characteristics | Total(n = 980) | ACC(n = 334) | NE(n = 437) | Other(n = 209) |
|---|---|---|---|---|
| **Age, years** | | | | |
| Median | 21 | 57 | 2 | 60 |
| IQR | 2–60 | 44–66 | 0–3 | 44–73 |
| **Race** | | | | |
| White | 763(77.9) | 276(82.6) | 328(75.1) | 159(76.1) |
| Black | 130(13.3) | 30(9.0) | 65(14.9) | 35(16.7) |
| Other | 83(8.5) | 26(7.8) | 43(9.8) | 14(6.7) |
| Unknown | 4(0.4) | 2(0.6) | 1(0.2) | 1(0.5) |
| **Gender** | | | | |
| Male | 493(50.3) | 135 (40.4) | 245(56.1) | 113(54.1) |
| Female | 487(49.7) | 199 (59.6) | 192(43.9) | 96(45.9) |
| **Years of diagnosis** | | | | |
| 2010 | 134(13.7) | 42(12.6) | 60(13.7) | 32(15.3) |
| 2011 | 112(11.4) | 40(12.0) | 45(10.3) | 27(12.9) |
| 2012 | 105(10.7) | 43(12.9) | 48(11.0) | 14(6.7) |
| 2013 | 127(13.0) | 41(12.3) | 62(14.2) | 24(11.5) |
| 2014 | 137(14.0) | 40(12.0) | 66(15.1) | 31(14.8) |
| 2015 | 126(12.9) | 47(14.1) | 56(12.8) | 23(11.0) |
| 2016 | 116(11.8) | 39(11.7) | 48(11.0) | 29(13.9) |
| 2017 | 123(12.6) | 42(12.6) | 52(11.9) | 29(13.9) |
| **Laterality** | | | | |
| Left | 485(49.5) | 157(47.0) | 237(54.2) | 91(43.5) |
| Right | 428(43.7) | 159(47.6) | 182(41.6) | 87(41.6) |
| Bilateral, single primary | 15(1.5) | 2(0.6) | 9(2.1) | 4(1.9) |
| Unknown | 52(5.3) | 16(4.8) | 9(2.1) | 18(12.9) |
| **Grade** | | | | |
| Well | 9(0.9) | 6(1.8) | 2(0.5) | 1(0.5) |
| Moderately | 13(1.3) | 11(3.3) | 0(0.0) | 2(1.0) |
| Poorly | 303(30.9) | 29(8.7) | 242(55.4) | 32(15.3) |
| Undifferentiated | 56(5.7) | 14(4.2) | 34(7.8) | 8(3.8) |
| Unknown | 599(61.1) | 274(82.0) | 159(36.4) | 166(79.4) |
| **Tumor size** | | | | |
| ≤100mm | 473(48.3) | 123(36.8) | 251(57.4) | 99(47.4) |
| >100mm, ≤200mm | 291(29.7) | 143(42.8) | 96(22.0) | 52(24.9) |
| >200mm | 29(3.0) | 26(7.8) | 2(0.5) | 1(0.5) |
| Unknown | 187(19.1) | 42(12.6) | 88(20.1) | 57(27.3) |
| **Median income** | | | | |
| <35,000 | 22(2.2) | 7(2.1) | 9(2.1) | 6(2.9) |
| 35,000–75,000 | 704(71.8) | 229(68.6) | 319(73.0) | 156(74.6) |
| >75,000 | 254(25.9) | 98(29.3) | 109(24.9) | 47(22.5) |
| **Residential areas** | | | | |
| Metropolis | 883(90.1) | 298(89.2) | 402(92.0) | 183(87.6) |
| Nonmetropolis | 92(9.4) | 36(10.8) | 32(7.3) | 24(11.5) |
| Unknown | 5(0.5) | 0(0.0) | 3(0.7) | 2(1.0) |
| **Surgery** | | | | |
| Yes | 534(54.5) | 123(36.8) | 343(78.5) | 68(32.5) |
| No | 442(45.2) | 209(62.6 | 93(21.3) | 140(67.0) |

(*Continued*)

**Table 1.** (Continued)

| Characteristics | Total(n = 980) | ACC(n = 334) | NE(n = 437) | Other(n = 209) |
|---|---|---|---|---|
| Unknown | 4(0.4) | 2(0.6) | 1(0.2) | 1(0.5) |
| **Metastasis site** | | | | |
| Only Bone | 236(45.9) | 18(11.5) | 185(71.7) | 33(33.3) |
| Only Brain | 7(1.4) | 4(2.5) | 1(0.4) | 2(2.0) |
| Only Liver | 153(29.8) | 58(36.9) | 68(26.4) | 27(27.3) |
| Only Lung | 118(23.0) | 77(49.0) | 4(1.6) | 37(37.4) |

**Abbreviations**: ACC, adrenal cortical carcinoma; NE, neuroblastoma.

The remarkable finding was that patients with NE were younger than those with the other types (P < 0.001).

## Distribution of metastasis pattern

The distributions of these patients in the sites of metastases of adrenal malignancy were presented in the Venn diagram (Fig 1). At the time of diagnosis, the most common site of

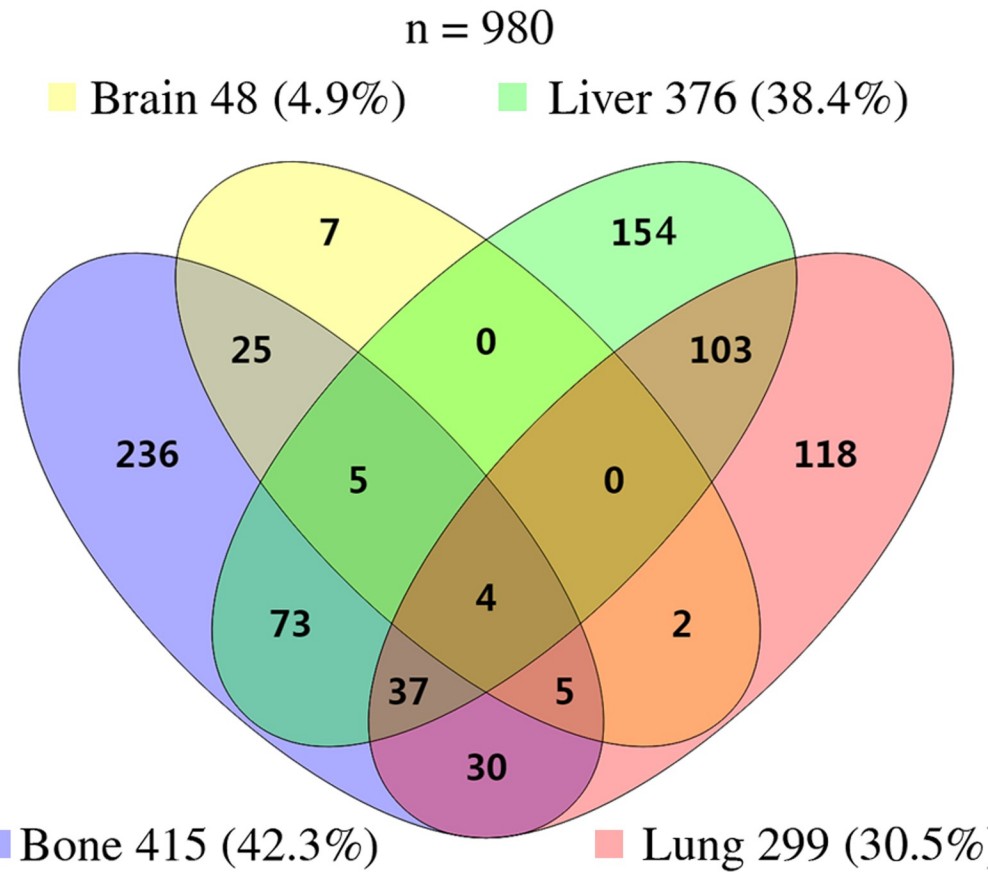

**Fig 1. Venn diagram of the distribution of distant metastatic sites.** There were four types of metastatic sites in 980 patients. Bone metastasis was the most common forms of metastasis.

metastases was bone (415 cases, 42.3%), followed by 376 (38.4%) patients with liver metastasis, 299 (30.5%) patients with lung metastasis, and 48 (4.9%) patients with brain metastasis. Patients who had a single site of distant metastases accounted for 52.6% (515/980), followed by two sites (233/980, 23.8%), three sites (47/980, 4.8%), and four sites (4/980, 0.4%). The clinical characteristics of these patients are presented (Table 2). The distribution of gender among the patients with bone metastasis and without bone metastasis was significantly different (P < 0.001). Depending on the histological type, the distribution was significantly associated with all four types of metastasis (all, P < 0.001). As for surgery, the distribution of patients was statistically significant except brain metastasis (P < 0.05).

The metastatic pattern of metastatic adrenal malignancy was exhibited in Table 3. Theoretically there were 15 metastatic forms, including 4 single metastases and 11 combinations of metastases. However, there were no relevant cases in this study in two types of metastatic forms, which were brain and liver metastases and brain and liver and lung metastases. We found bone metastasis was the most common metastasis in single metastatic patients (24.08%), followed by liver (15.71%), lung (12.04%) and brain (0.71%). As for two sites, the highest frequency was observed in patients with liver and lung metastases at 10.51% (103/980). Bone metastasis presented better survival rate in single metastasis. Patients with brain and lung metastases had the worse survival rate than other metastatic types in two sites metastases. In addition, median OS cannot be concluded in only bone metastasis and bone and brain metastases in Kaplan-Meier analysis.

## Metastatic sites and survival outcomes

In this study, 550 deaths (56.10%) were observed. The mortality rate was high. We performed Kaplan-Meier analysis in patients with a single metastatic site (n = 515). It was routinely observed that patients with bone metastases had better outcomes in OS and CSS, compared with patients diagnosed with liver, lung, and brain metastases. Log-rank test was used to compare the number of all metastatic sites of adrenal malignancy in patients, and it was found that the patients with one metastatic site had significantly longer OS and CSS than those with more metastases (OS: HR = 1.618, 95% CI = 1.339–1.955; CSS: HR = 1.680, 95% CI = 1.373–2.054; both P < 0.001) (Fig 2). Patients with bone and liver metastases presented longer OS and CSS compared to those with lung and brain metastases in the group of only one metastatic site by using the log-rank test (P < 0.001) (Fig 3).

## Univariate survival analysis

Prognostic factors, including the site of metastases, gender, primary site, histological types, and residential areas, were analyzed in the univariate survival analysis (Table 4). Patients with brain metastasis exhibited the worst OS and CSS than those with the other three types of single metastasis (OS: HR = 7.447, 95% CI = 3.416–16.233; CSS: HR = 7.830, 95% CI = 3.143–19.506). We observed that females exhibited worse prognosis than males in OS and CSS (P < 0.05). Compared with the tumor arising from the cortex, the tumor arising from the medulla showed better survival outcomes (P < 0.001). Similarly, patients with NE seemed to have better OS and CSS than those with ACC. And patients with surgery had better OS and CSS than those without (P < 0.001). Also, there was a significant difference between patients who lived in metropolis and nonmetropolis. Patients who lived in metropolis presented a better survival.

## Multivariable survival analysis

The parameters, including metastatic site, gender, histology, primary site, residential areas and surgery, were selected in multivariate analysis. As for metastatic site, brain metastasis was still

**Table 2. Clinical characteristics and metastasis sites.**

| Features | Bone metastasis (%) | | | Brain metastasis (%) | | | Liver metastasis (%) | | | Lung metastasis (%) | | |
|---|---|---|---|---|---|---|---|---|---|---|---|---|
| No | No | Yes | P | No | Yes | P | No | Yes | P | No | Yes | P |
| **Age** | | | | | | | | | | | | |
| Median | 51 | 3 | <0. | 22 | 3 | 0.0 | 7 | 39 | 0.0 | 4 | 53 | <0. |
| IQR | 4~64 | 2~16 | 001 | 2~59 | 2~13 | 02 | 2~58 | 1~59 | 01 | 1~54 | 29~65 | 001 |
| **Race** | | | | | | | | | | | | |
| White | 452(59.2) | 311(40.8) | >0.05[b] | 730(95.7) | 33(4.3) | >0.05[b] | 466(61.1) | 297(38.9) | >0.05[b] | 531(69.6) | 232(30.4) | >0.05[b] |
| Black | 62(47.7) | 68(52.3) | | 120(92.3) | 10(7.7) | | 85(65.4) | 45(34.6) | | 93(71.5) | 37(28.5) | |
| Other | 49(59.0) | 34(41.0) | | 78(94.0) | 5(6.0) | | 51(61.4) | 32(38.6) | | 54(65.1) | 29(34.9) | |
| Unknown | 2(50.0) | 2(66.7) | | 4(100.0) | 0(0.0) | | 2(50.0) | 2(50.0) | | 3(75.0) | 1(25.0) | |
| **Gender** | | | | | | | | | | | | |
| Male | 257(52.1) | 236(47.9) | <0.001[a] | 465(94.3) | 28(5.7) | >0.05[a] | 313(63.5) | 180(36.5) | >0.05[a] | 360(73.0) | 133(27.0) | <0.05[a] |
| Female | 308(63.2) | 179(36.8) | | 467(95.9) | 20(4.1) | | 291(59.8) | 196(40.2) | | 321(65.9) | 166(34.1) | |
| **Primary site** | | | | | | | | | | | | |
| Cortex | 153(80.5) | 37(19.5) | <0.001[a] | 186(97.9) | 4 (2.1) | <0.05[b] | 94(49.5) | 96(50.5) | <0.05[a] | 95(50.0) | 95(50.0) | <0.001[a] |
| Medulla | 16(47.1) | 18(52.9) | | 29(85.3) | 5(14.7) | | 20(58.8) | 14(41.2) | | 23(67.6) | 11(32.4) | |
| Adrenal gland, NOS | 396(52.4) | 360(47.6) | | 717(94.8) | 39(5.2) | | 490(64.8) | 266(35.2) | | 563(74.5) | 193(25.5) | |
| **Histologic type** | | | | | | | | | | | | |
| ACC | 276(82.6) | 58(17.4) | <0.001[a] | 329(98.5) | 5(1.5) | <0.001[a] | 165(49.4) | 169(50.6) | <0.001[a] | 159(47.6) | 175(52.4) | <0.001[a] |
| NE | 144(33.0) | 293(67.0) | | 403(92.2) | 34(7.8) | | 300(68.6) | 137(31.4) | | 395(90.4) | 42(9.6) | |
| Other types | 145(69.4) | 64(30.6) | | 200(95.7) | 9(4.3) | | 139(66.5) | 70(33.5) | | 127(60.8) | 82(39.2) | |
| **Laterality** | | | | | | | | | | | | |
| Left | 260(53.9) | 225(46.4) | <0.05[b] | 454(93.6) | 31(6.4) | >0.05[b] | 296(61.0) | 189(39.0) | >0.05[a] | 336(69.3) | 149(30.9) | >0.05[b] |
| Right | 255(59.6) | 173(40.4) | | 414(96.7) | 14(3.3) | | 270(63.1) | 158(36.9) | | 301(70.3) | 127(29.7) | |
| Bilateral, single primary | 11(73.3) | 4(26.7) | | 14(93.3) | 1(6.7) | | 6(40.0) | 9(60.0) | | 12(80.0) | 3(20.0) | |
| Unknown | 39(75.0) | 13(25.0) | | 50(96.2) | 2(3.8) | | 32(61.5) | 20(38.5) | | 32(61.5) | 20(38.5) | |
| **Grade** | | | | | | | | | | | | |
| Well | 4(44.4) | 5(55.6) | <0.001[b] | 1(11.1) | 8(88.9) | <0.05[b] | 4(44.4) | 5(55.6) | <0.05[b] | 4(44.4) | 5(55.6) | <0.001[b] |
| Moderately | 3(23.1) | 10(76.9) | | 1(7.7) | 12(92.3) | | 2(15.4) | 11(84.6) | | 5(38.5) | 8(61.5) | |
| Poorly | 174(57.4) | 129(42.6) | | 24(7.9) | 279(92.1) | | 99(32.7) | 204(67.3) | | 48(15.8) | 255(84.2) | |
| Undifferentiated | 28(50.0) | 28(50.0) | | 3(5.4) | 53(94.6) | | 18(32.1) | 38(67.9) | | 11(19.6) | 45(80.4) | |
| Unknown | 206(34.4) | 393(65.6) | | 19(3.2) | 580(96.8) | | 253(42.2) | 346(57.8) | | 231(38.6) | 368(61.4) | |
| **Tumor size** | | | | | | | | | | | | |
| ≤100mm | 237(50.1) | 236(49.9) | <0.001[b] | 34(7.2) | 439(92.8) | <0.05[b] | 170(35.9) | 303(64.1) | >0.05[a] | 116(24.5) | 357(75.5) | <0.001[a] |
| >100mm, ≤≤200mm | 99(34.0) | 192(66.0) | | 7(2.4) | 284(97.6) | | 116(39.9) | 175(60.1) | | 122(41.9) | 169(58.1) | |
| >200mm | 3(10.3) | 26(89.7) | | 0(0.0) | 29(100.0) | | 15(51.7) | 14(48.3) | | 11(37.9) | 18(62.1) | |
| **Median income** | | | | | | | | | | | | |
| <35,000 | 15(68.2) | 7(31.8) | >0.05[a] | 19(86.4) | 3(13.6) | >0.05[b] | 15(68.2) | 7(31.8) | >0.05[a] | 17(77.3) | 5(22.7) | >0.05[a] |
| 35,000–75,000 | 394(56.0) | 310(44.0) | | 668(94.9) | 36(5.1) | | 439(62.4) | 265(37.6) | | 489(69.5) | 215(30.5) | |
| >75,000 | 156(61.4) | 98(38.6) | | 245(96.5) | 9(3.5) | | 150(59.1) | 104(40.9) | | 175(68.9) | 79(31.1) | |
| **Residential areas** | | | | | | | | | | | | |
| Metropolis | 501(56.7) | 382(43.3) | >0.05[b] | 840(95.1) | 43(4.9) | >0.05[b] | 548(62.1) | 335(37.9) | >0.05[b] | 617(69.9) | 266(30.1) | >0.05[b] |
| Nonmetropolis | 60(65.2) | 32(34.8) | | 87(94.6) | 5(5.4) | | 53(57.6) | 39(42.4) | | 60(65.2) | 32(34.8) | |
| Unknown | 4(80.0) | 1(20.0) | | 5(100.0) | 0(0.0) | | 3(60.0) | 2(40.0) | | 4(80.0) | 1(20.0) | |
| **Surgery** | | | | | | | | | | | | |
| Yes | 295(55.2) | 239(44.8) | <0.001[a] | 30(5.6) | 504(94.4) | >0.05[a] | 151(28.3) | 383(71.7) | <0.001[a] | 100(18.7) | 434(81.3) | <0.001[a] |

(*Continued*)

**Table 2.** (Continued)

| Features | Bone metastasis (%) | | | Brain metastasis (%) | | | Liver metastasis (%) | | | Lung metastasis (%) | | |
|---|---|---|---|---|---|---|---|---|---|---|---|---|
| No | No | Yes | P | No | Yes | P | No | Yes | P | No | Yes | P |
| No | 119(26.9) | 323(73.1) | | 18(4.1) | 424(95.9) | | 224(50.7) | 218(49.3) | | 199(45.0) | 243(55.0) | |

[a]. The P value was obtained from Pearson chi-Square test.

[b]. The P value was obtained from Fisher's exact test.

**Abbreviations**: NOS, not otherwise specified; ACC, adrenal cortical carcinoma; NE, neuroblastoma.

the worst prognostic metastasis both in OS and CSS. The same as the univariate analysis, primary site was an independent prognostic factor for adrenal metastatic cancer patients both in OS and CSS (P < 0.05). In the analysis of the primary site, the primary site in the cortex had a worse prognosis than that in the medulla in both OS and CSS (P < 0.05). With respect to the surgery, patients underwent the surgery showed the better prognosis for OS and CSS (all P < 0.001) (Table 5).

## Nomogram

A nomogram was established for OS based on the multivariate analysis results and data availability (Fig 4). We could predict the 1, 3, and 5 year survival probability in the base of the sum of the dot scale at the top and the points for every factor. The probabilities of 1, 3, and 5 year survival relied on the fractional proportion present at the bottom of the nomogram. The C-

**Table 3. Frequencies of combination metastasis and 3, 5-y OS.**

| | Number (%) | 3-y OS | 5-y OS | Median OS (months) |
|---|---|---|---|---|
| **One site** | | | | |
| Only Bone | 236 (24.08) | 73.31% | 67.80% | NA |
| Only Brain | 7 (0.71) | 0.00% | 0.00% | 2.0 |
| Only Liver | 154 (15.71) | 48.05% | 45.45% | 15.0 |
| Only Lung | 118 (12.04) | 25.42% | 22.89% | 9.0 |
| **Two sites** | | | | |
| Bone and brain | 25 (2.55) | 80.00% | 72.00% | NA |
| Bone and liver | 73 (7.45) | 54.79% | 53.42% | 41.0 |
| Bone and lung | 30 (3.06) | 40.00% | 36.67% | 10.0 |
| Brain and liver | 0 (0.00) | 0 | 0 | / |
| Brain and lung | 2 (0.20) | 0 | 0 | 1.5 |
| Liver and lung | 103 (10.51) | 22.33% | 20.39% | 3.0 |
| **Three sites** | | | | |
| Bone and brain and liver | 5 (0.51) | 40.00% | 40.00% | 20.5 |
| Bone and brain and lung | 5 (0.51) | 20.00% | 20.00% | 19.0 |
| Bone and liver and lung | 37 (3.78) | 24.32% | 21.62% | 7.0 |
| Brain and liver and lung | 0 (0.00) | 0 | 0 | / |
| **Four sites** | | | | |
| Bone and brain and liver and lung | 4 (0.41) | 50.00% | 50.00% | 3.0 |

NA: The median OS cannot be concluded because more than half of the patients still alive at the end of the follow up; /: The median OS cannot be calculated due to the absence of related cases.

**Abbreviations**: OS: overall survival.

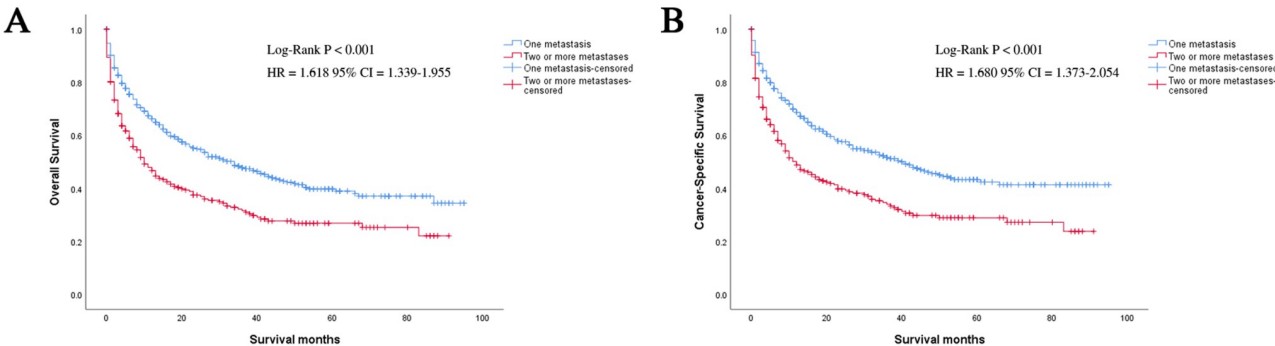

**Fig 2. Kaplan-Meier curves of OS and CSS according to the number of metastatic sites.** Patients in one metastasis had significantly better survival both in OS and CSS; (a) OS; (b) CSS; OS, overall survival; CSS, cancer-specific survival; HR, hazard ratio; CI, confidence interval.

index was 0.76, which means the nomogram is a reliable model for predicting the OS of metastatic adrenal malignancy.

## Discussion

Primary adrenal malignancy is rare, and metastatic adrenal malignancy is even rarer in clinical practice. Even in the SEER database, patients with primary metastatic adrenal malignancy are relatively less in number compared to patients with other malignancies. SEER breaks the barrier of minor cases series and isolated institutional studies and provides a platform for learning rare tumor deeply. With respect to the adrenal tumor, the only standard to indicate malignancy is the existence of metastasis [14]. In this study, we analyzed primary adrenal malignancy with distant sites of metastasis, including bone, brain, liver, and lung, as the recorded sites in the SEER database after 2010. At the time of diagnosis, the rate of metastasis to bone, liver, lung, and brain in adrenal malignancy patients was 42.3%, 38.4%, 30.5%, and 4.9%, respectively. There were plenty of articles about adrenal incidentaloma and adrenal metastasis secondary to other diseases [5, 6, 15], but relevant literature reports about primary metastatic adrenal malignancy were limited, which meant that few studies highlighted the relation between metastatic sites and survival. Our study was the first to explore the role of the metastatic site in adrenal malignancy patients' survival using a large sample size. Besides,

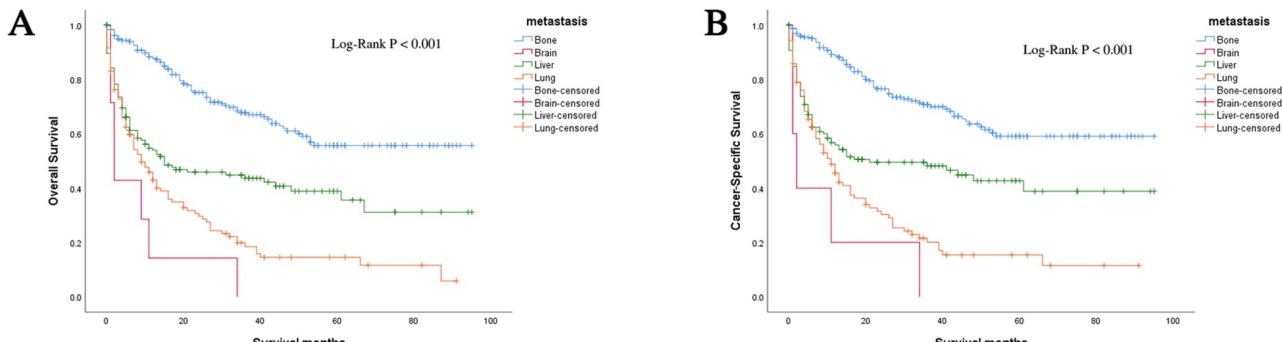

**Fig 3. Kaplan-Meier curves and log-rank test for OS and CSS according to the site of metastasis (only one site).** Patients in bone metastasis had the best survival outcomes in OS and CSS. Brain metastasis had significantly shorter OS and CSS compared with other types of sites; (a) OS; (b) CSS; OS, overall survival; CSS, cancer-specific survival; HR, hazard ratio; CI, confidence interval.

**Table 4. Univariate survival analysis predicting overall survival and cancer-specific survival in patients with four single metastases.**

| Risk Factors | Overall survival | | Cancer-specific survival | |
|---|---|---|---|---|
| | HR(95%CI) | P | HR(95%CI) | P |
| **Metastasis site** | | <0.001 | | <0.001 |
| Only Bone | Ref. | | Ref. | |
| Only Brain | 7.447(3.416–16.233) | <0.001 | 7.830(3.143–19.506) | <0.001 |
| Only Liver | 2.527(1.853–3.447) | <0.001 | 2.499(1.788–3.492) | <0.001 |
| Only Lung | 4.128(3.037–5.610) | <0.001 | 3.069–5.923 | <0.001 |
| **Gender** | | 0.016 | | 0.007 |
| Male | Ref. | | Ref. | |
| Female | 1.229(1.039–1.453) | 0.016 | 1.280(1.071–1.531) | 0.007 |
| **Primary site** | | <0.001 | | <0.001 |
| Cortex | Ref. | | Ref | |
| Medulla | 0.379(0.224–0.640) | <0.001 | 0.312(0.168–0.580) | <0.001 |
| **Histology** | | <0.001 | | <0.001 |
| **ACC** | Ref. | | Ref. | |
| **NE** | 0.169(0.136–0.210) | <0.001 | 0.162(0.129–0.204) | <0.001 |
| **Other types** | 0.865(0.710–1.054) | 0.150 | 0.819(0.661–1.016) | 0.069 |
| **Residential areas** | | <0.001 | | <0.001 |
| Metropolis | Ref | | Ref. | |
| Nonmetropolis | 1.667(1.288–2.158) | <0.001 | 1.729(1.313–2.276) | <0.001 |
| Unknown | 1.570(0.587–4.200) | 0.369 | 1.429(0.459–4.450) | 0.538 |
| **Surgery** | | <0.001 | | <0.001 |
| Yes | Ref. | | Ref. | |
| No | 3.307(2.779–3.936) | <0.001 | 3.263(2.713–3.923) | <0.001 |

**Abbreviations**: HR, hazard ratio; CI, confidence interval; Ref, reference; ACC, adrenal cortical carcinoma; NE, neuroblastoma.

information on site-specific survival was also available to learn about the prognosis of metastatic adrenal malignancy in more detail.

Traditional studies about the incidence of ACC presented a bimodal age distribution in the first and fourth decades of life, which was in contrast with our study [16]. Interestingly, the result of the previous study based on the SEER registries and international registries was consistent with our study, and this discrepancy might be attributed to the differences in the time of diagnosis. To date, ACC has shown insignificant changes in survival outcomes [17–20]. In contrast, NE appears to show more preferable oncological outcomes with drastic improvements in treatment, and the 1-year CSS rate of NE has exceeded 90% [2]. The same phenomenon was also observed in our study. Metastatic NE had better oncological survival than metastatic ACC. Therefore, treatments of ACC still have a long way to go. Although there was a report about the diameter of the tumor being associated with the risk of malignancy [21], the SEER data included was not applicable for evaluating the risk, and over the half of the patients had incomplete data on tumor size.

In our analysis, the NE subtype had a significantly higher rate of bone metastasis (67%) while the ACC type was more likely to have lung (52.4%) and liver (50.6%) metastases. Males were more likely to have bone metastasis, and females were prone to develop liver and lung metastases. It makes sense since NE tends to affect males and ACC prefers the female sex, and that is why females exhibited worse prognosis than males in both OS and CSS. Irrespective of the population of adults or children with ACC, the ratio of females to males ranged from 1.5–

**Table 5. Multivariate survival analysis predicting overall survival and cancer-specific survival in patients with four single metastases.**

| Characteristics | Overall survival | | Cancer-specific survival | |
|---|---|---|---|---|
| | HR (95% CI) | P | HR (95% CI) | P |
| **Metastasis site** | | | | |
| **Only bone** | Ref. | | Ref. | |
| **Only brain** | 11.133(2.074–59.757) | 0.005 | 13.984(2.496–78.334) | 0.003 |
| **Only liver** | 1.144(0.477–2.746) | 0.763 | 1.458(0.571–3.724) | 0.431 |
| **Only lung** | 1.228(0.543–2.776) | 0.621 | 1.335(0.558–3.191) | 0.516 |
| **Gender** | | | | |
| Male | Ref. | | Ref. | |
| Female | 0.904(0.526–1.554) | 0.715 | 0.862(0.479–1.552) | 0.621 |
| **Primary site** | | | | |
| Cortex | Ref. | | Ref. | |
| Medulla | 0.218(0.068–0.692) | 0.010 | 0.181(0.048–0.678) | 0.011 |
| **Histology** | | | | |
| ACC | Ref. | | Ref | |
| NE | 0.306(0.066–1.424) | 0.131 | 0.287(0.061–1.355) | 0.115 |
| Others | 1.196(0.493–2.898) | 0.692 | 1.144(0.466–2.808 | 0.770 |
| **Residential areas** | | | | |
| Metropolis | Ref | | Ref | |
| Nonmetropolis | 1.840(0.889–3.808) | 0.100 | 2.121(0.930–4.838) | 0.074 |
| **Surgery** | | | | |
| Yes | Ref. | | Ref. | |
| No | 4.861(2.449–9.647) | <0.001 | 4.826(2.386–9.761) | <0.001 |

**Abbreviations:** HR, hazard ratio; CI, confidence interval; Ref, reference; ACC, adrenocortical carcinoma; NE, neuroblastoma.

2.5:1 [22, 23]. Females exhibited worse HR in univariate analysis because of the inferior prognosis and large population of ACC. It was interesting to observe that the lesion on the left side was prone to bone metastasis, and we also found that NE was related to laterality and the number of lesions on the left side was more than that on the right side. The same occurrence was reported in previous studies of ACC; the left side was more and easier to be affected [24–26]. Nevertheless, our study did not display this in ACC; may be the laterality was influenced by the patients. Besides, we noted that race, median household income, and residential areas were not associated with metastatic sites.

In survival analysis, we also obtained some findings. First, we found that patients with brain metastasis presented the worst OS and CSS in both univariate and multivariate survival analyses. The most common subtype of brain metastasis was NE. Some reports have described the potential relation between NE and brain [27, 28], but the explanation of this phenomenon remains obscure; more basic research is needed to explore the biological mechanisms. The primary site also showed significance in OS and CSS, even in multivariate analysis. Considering the primary site of subtype, we did not include this item in the nomogram because the primary sites of ACC were from the cortex.

To our knowledge, this is the first SEER-based study focusing solely on the hematogenous metastatic pattern of adrenal malignancy patients. Nevertheless, there are several limitations due to the retrospective nature of the study. At first, the database only provides information on synchronous metastasis to bone, brain, liver, and lung from 2010 and the follow-up time is not very long. Compared with metachronous metastasis, the patients of our study was relatively

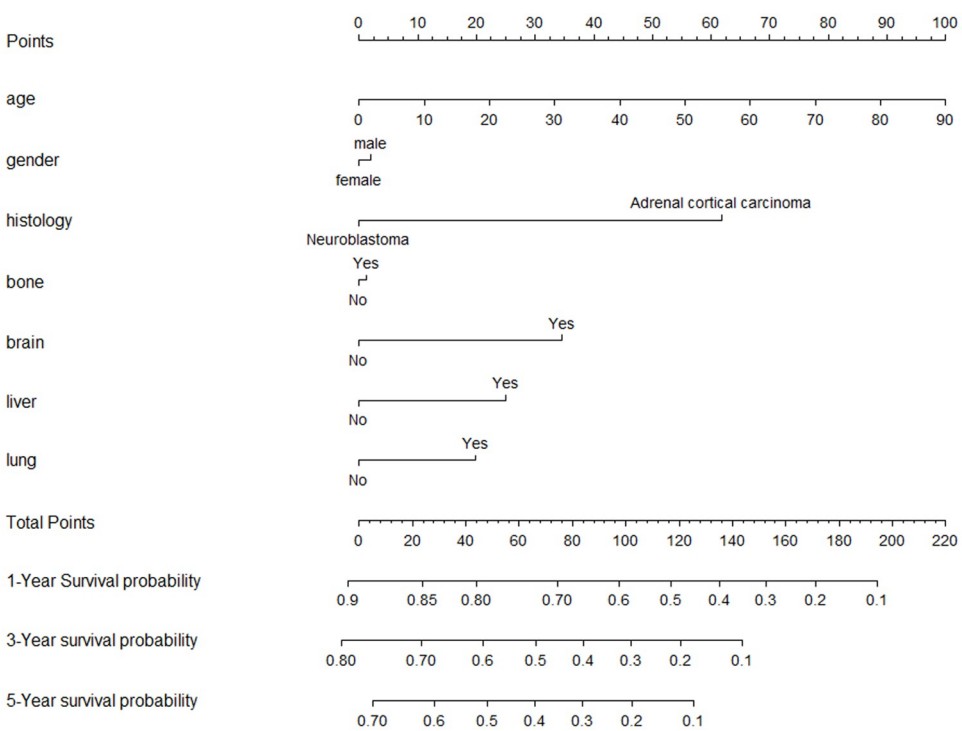

**Fig 4. Nomogram for predicting the 1-year, 3-year and 5-year overall survival in patients with primary metastatic adrenal malignancy.** To obtain the predicted survival probability in meters, locate patient values on each axis. Based on the points line to acquire the number of points to add. Sum the points of all variables to determine the total point. A vertical line can be drawn down to the 1-year, 3-year and 5-year overall survival probability.

minor. Second, the lack of information regarding treatment regimens or surgery for the included patients might give rise to potential confounders.

## Conclusion

In conclusion, heterogeneity exists in the oncological outcomes of patients with site-specific metastasis. Patients with bone metastasis appear to show the best oncologic survival and those with brain metastasis show the worst survival among those with single metastasis. The nomogram may help predict the 1-year, 3-year and 5-year overall survival in patients with primary metastatic adrenal malignancy. Relying on different histological types, there are numerous metastatic features and prognostic values. Knowledge of these differences in metastatic patterns and pathological types may contribute to designing targeted pre-treatment assessment of adrenal malignancy and making a personalized curative intervention. Further efforts are still need to investigate the relationship between more comprehensive factors and adrenal malignancy in the future.

## Author Contributions

**Conceptualization:** Haibin Wei, Dahong Zhang.

**Data curation:** Jia Miao, Feng Liu.

**Formal analysis:** Jianxin Cui.

**Funding acquisition:** Haibin Wei.

**Methodology:** Jianxin Cui, Qi Zhang, Zujie Mao.

**Resources:** Qi Zhang.

**Software:** Qi Zhang, Zujie Mao.

**Supervision:** Haibin Wei, Dahong Zhang.

**Validation:** Jianxin Cui, Qi Zhang, Zujie Mao.

**Visualization:** Jia Miao, Feng Liu.

**Writing – original draft:** Jia Miao.

**Writing – review & editing:** Haibin Wei, Dahong Zhang.

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
