## [Decision Letter · Decision Letter 0]

19 Aug 2021

PONE-D-21-11886

The prognosis of different distant metastases pattern in malignant tumors of the adrenal glands: A population-based retrospective study

PLOS ONE

Dear Dr. Zhang,

Thank you for submitting your manuscript to PLOS ONE. After careful consideration, we feel that it has merit but does not fully meet PLOS ONE’s publication criteria as it currently stands. Therefore, we invite you to submit a revised version of the manuscript that addresses the points raised during the review process.

We look forward to receiving your revised manuscript.

Kind regards,

Filomena de Nigris, M.D., Ph.D.

Academic Editor

PLOS ONE

Journal Requirements:

Reviewers' comments:

Reviewer's Responses to Questions

**Comments to the Author**

1. Is the manuscript technically sound, and do the data support the conclusions?

Reviewer #1: Partly

Reviewer #2: Partly

Reviewer #3: Yes

2. Has the statistical analysis been performed appropriately and rigorously? 

Reviewer #1: No

Reviewer #2: Yes

Reviewer #3: Yes

3. Have the authors made all data underlying the findings in their manuscript fully available?

Reviewer #1: Yes

Reviewer #2: Yes

Reviewer #3: Yes

4. Is the manuscript presented in an intelligible fashion and written in standard English?

Reviewer #1: Yes

Reviewer #2: Yes

Reviewer #3: Yes

5. Review Comments to the Author

Reviewer #1: Dear Editor, Zhang the interesting paper by ..et al the aims to investigate the impact of different 24 distant metastases pattern on the survival of patients with adrenal malignancy.

Major finds comes from the fact that many confounding factors are not evaluated such as tumor size, treatment regime, surgery, grade of tumor, so these events influence the follow up of patient and prognosis.

Another major point is that in lane 169 the authors indicated that site of metastases was an independent prognostic factor (P <0.05) from the table 4 seams that brain metastasis were statistically associated with prognosis

moreover in the presentation of data the authors should better indicate numerically if one condition improves the survival or prognosis . In tables are not indicated how many patients had one or more metastasis reported instead in Kaplan analysis

Minor points

The Ven diagramm needs a color legend

The quality of image is poor

Reviewer #2: The main points that the authors should clarify are the following:

1) what is the percentage of metastatic cancer at the time of diagnosis

2) how many patients have more than one metastasis

3) since it is not clear what the median survival range of subjects presenting metastases to bone versus brain is, the authors should better comment on the Kaplan-Meier analysis and indicate in the text which was the best survival with several metastatic sites

4) in the conclusions section, authors should more clearly describe whether knowledge of these differences in metastatic patterns could help better guide pre-treatment evaluation of cancer and determine interventions with curative intent.

Minor points

1) Authors should submit a figure legend for Figure 1 Figure 1

2) Authors should provide image with higher resolution, since image quality is poor

Reviewer #3: Dear Editor

In the present study, Zhang.et al performed several experiments aimed to prove the impact of different 24 distant metastases pattern on the survival of patients with adrenal malignancy.

The work is well elaborated, and the focus is very interesting. However, I find some parts of the article need to be improved and I have some revisions to add before acceptance. I recommend acceptance after revision.

Major finds

1) The authors should improve the clinical data added the percentage of metastatic cancer at diagnosis.

2) The authors should clarify which is the mean survival of metastases with bone compare to brain

3) The author should improve the comment about the “Kaplan analysis” and highlight which was the best survival with different metastatic sites in the text

4) In the "Conclusion" section the authors should better explain the impact of their work

Minor points

The authors should improve the quality of images

The “Ven diagram” needs a color legend

6. PLOS authors have the option to publish the peer review history of their article (what does this mean?). If published, this will include your full peer review and any attached files.

Reviewer #1: No

Reviewer #2: **Yes: **Concetta Schiano

Reviewer #3: No

---

## [Author Response · Author response to Decision Letter 0]

4 Sep 2021

Editorial Office, 

PLOS ONE September 3rd, 2021

Dear Prof. Filomena de Nigris,

Thank you for your letter and advice regarding the review and revision of our manuscript entitled “The prognosis of different distant metastases pattern in malignant tumors of the adrenal glands: A population-based retrospective study”. The submission ID is PONE-D-21-11886. We have addressed all of the comments raised by the reviewers and have revised the paper accordingly. The amendments are highlighted in the revised manuscript. A point-to-point reply is included as follows. We are grateful to the high appreciations and constructive suggestions on our manuscript from you and three reviewers.

We would like to re-submit the revised manuscript for your consideration. We hope that the revision is acceptable, and we look forward to hearing from you soon.

Best Regards!

Yours sincerely,

Haibin Wei MD, and Da-hong Zhang, M.D. 

1. Haibin Wei, Department of Urology, Zhejiang Provincial People's Hospital, People's Hospital of Hangzhou Medical College, No. 158, Shangtang Road, Xiacheng District, Hangzhou 310014, Zhejiang, China. Tel. +86 137 7786 0207. Fax: +86 571 85131448. Email address: whb-sysu@163.com.

2. Dahong Zhang, Department of Urology, Zhejiang Provincial People's Hospital, People's Hospital of Hangzhou Medical College, No. 158, Shangtang Road, Xiacheng District, Hangzhou 310014, Zhejiang, China. Tel. +86 571 8589 3312. Fax: +86 571 85131448. ORCID: 0000-0002-6934-7956. Email address: zhangdahong88@yeah.net.

We would like to express our sincere gratitude to the reviewers for their constructive and positive comments.

Reviewer #1: 

Dear Editor, Zhang the interesting paper by ..et al the aims to investigate the impact of different 24 distant metastases pattern on the survival of patients with adrenal malignancy.

1. Major finds comes from the fact that many confounding factors are not evaluated such as tumor size, treatment regime, surgery, grade of tumor, so these events influence the follow up of patient and prognosis. 

Response: The reviewer’s comment is well appreciated. Thank you for the reminder. 

We do agree with you that some factors influence the follow up of patient and prognosis, such as tumor size, treatment regime, surgery, grade of tumor.

First, according to your comment, we added the variable “surgery” and defined the variable as the corresponding definition of the SEER database. As for table 2, we added “As for surgery, the distribution of patients was statistically significant except brain metastasis (P < 0.05).” to the corresponding section of the revised manuscript. As for univariate analysis, we added “And patients with surgery had better OS and CSS than those without (P < 0.001).”

Second, we have added new variables like tumor size and grade of tumor in table 1 and table 2 and made the corresponding changes in all relevant section of the manuscript. For table1, we added the corresponding description as “In addition to the grade of unknown, the most common grade was poor differentiated (30.9%).” and “48.3% of patients had tumor size �100mm, but 19.1% had unknown tumor size.” in the revised manuscript.

Third, although “SEER research data, 18 registries, Nov 2019 Sub (2000- 2017)” database has different variables and a larger sample size than other database, some of the data is incomplete, such as pathological grade. For the grade of tumor, the data for 82 % of the adrenal cortical carcinoma and 36.4% of the neuroblastoma were unknown, respectively. So we did not include it in univariate survival analysis to avoid the bias of incomplete data.

Fourth, this study focuses on evaluating prognosis of different distant metastases pattern in adrenal malignancy patients at the first diagnosis. We aimed to provide a better understanding of the prognosis before diverse treatments. Therefore, the other treatment strategies were not included in the study.

Thank you very much for your great comment.

2. Another major point is that in lane 169 the authors indicated that site of metastases was an independent prognostic factor (P <0.05) from the table 4 seams that brain metastasis were statistically associated with prognosis

Response: The reviewer’s comment is well appreciated. Thank you for your reminding. According to your comment, we have changed the sentence into “The parameters, including metastatic site, gender, histology, primary site, residential areas and surgery, were selected in multivariate analysis. As for metastatic site, brain metastasis was still the worst prognostic metastasis both in OS and CSS. The same as the univariate analysis, primary site was an independent prognostic factor for adrenal metastatic cancer patients both in OS and CSS (P < 0.05).” in the revised manuscript.

3. moreover in the presentation of data the authors should better indicate numerically if one condition improves the survival or prognosis. 

Response: The reviewer’s comment is well appreciated. Thank you for your valuable advice.

First, we have added a new table titled “Frequencies of combination metastasis and 3, 5-y OS” in the revised manuscript. In this table, the 3, 5-y overall survival and median OS were shown numerically.

Second, in our study, with respect to the surgery, patients (suffering from a single metastasis) underwent the surgery showed the better prognosis for OS and CSS in univariate survival analysis and multivariable survival analysis. We do agree with you that specific digitization may be a good choice if some condition improves the survival or prognosis. This study focuses on prognosis of different distant metastases pattern, not on therapeutic intervention for metastatic adrenal malignancy patients. The optimization of intervention conditions will be an important part of subsequent research.

Thanks for your suggestion.

4. In tables are not indicated how many patients had one or more metastasis reported instead in Kaplan analysis. 

Response: The reviewer’s comment is well appreciated. Thank you for your reminding. We have added a new table about the frequency of one or more metastasis. 

First, the number of one site, two sites, three sites, and four sites are shown in Table 3, respectively.

Second, according to your comment, we have added “The metastatic pattern of metastatic adrenal malignancy was exhibited in Table 3. Theoretically there were 15 metastatic forms, including 4 single metastases and 11 combinations of metastases. However, there were no relevant cases in this study in two types of metastatic forms, which were brain and liver metastases and brain and liver and lung metastases. We found bone metastasis was the most common metastasis in single metastatic patients (24.08%), followed by liver (15.71%), lung (12.04%) and brain (0.71%). As for two sites, the highest frequency was observed in patients with liver and lung metastases at 10.51% (103/980). Lung metastasis presented better early survival rate in single metastasis. Patients with brain and lung metastases had the worse survival rate than other metastatic types in two sites metastases” in the revised manuscript.

Thank you for your suggestion.

5. Minor points

The Ven diagramm needs a color legend

The quality of image is poor

Response: The reviewer’s comment is well appreciated. We have modified the image according to journal’s requirements. We are grateful to the constructive suggestions on our manuscript from you.

Reviewer #2: 

The main points that the authors should clarify are the following:

1. what is the percentage of metastatic cancer at the time of diagnosis. 

Response: Thank you very much for your comment. According to your comment, we have added “A total of 2053 eligible patients with primary adrenal malignancy between 2010 and 2017 were identified from SEER database. Among these patients, the metastatic adrenal malignancy accounted for 47.74 %( 980/2053) at the time of diagnosis, including 493 men (50.3%) and 487 women (49.7%). The metastatic information on bone, brain, liver, and lung metastasis was collected in the SEER database." in the revised manuscript.

2. how many patients have more than one metastasis.

Response: The reviewer’s comment is appreciated. We added a new table about the frequency of four single metastases and 9 combinations of metastases. The number of one site, two sites, three sites, and four sites are shown in Table 3, respectively.

3. since it is not clear what the median survival range of subjects presenting metastases to bone versus brain is, the authors should better comment on the Kaplan-Meier analysis and indicate in the text which was the best survival with several metastatic sites

Response: Thank you for your reminding. We have added a new table about the median survival and the frequency of combination metastasis in this manuscript (Table 3). And we added “The metastatic pattern of metastatic adrenal malignancy was exhibited in Table 3. Theoretically there were 15 metastatic forms, including 4 single metastases and 11 combinations of metastases. However, there were no relevant cases in this study in two types of metastatic forms, which were brain and liver metastases and brain and liver and lung metastases. We found bone metastasis was the most common metastasis in single metastatic patients (24.08%), followed by liver (15.71%), lung (12.04%) and brain (0.71%). As for two sites, the highest frequency was observed in patients with liver and lung metastases at 10.51% (103/980). Lung metastasis presented better early survival rate in single metastasis Patients with brain and lung metastases had the worse survival rate than other metastatic types in two sites metastases. In addition, median OS cannot be concluded in only bone metastasis and bone and brain metastases in Kaplan-Meier analysis.” in the revised manuscript.

4. in the conclusions section, authors should more clearly describe whether knowledge of these differences in metastatic patterns could help better guide pre-treatment evaluation of cancer and determine interventions with curative intent.

Response: 

The reviewer’s comment is well appreciated. Thank you for your constructive suggestion. We do agree with you that it is very necessary to clearly articulate the significance of metastatic patterns for pre-treatment evaluation and clinical interventions.

First, various prognostic values are associated with different metastatic sites, and knowledge of overall survival is very important for pre-treatment evaluation. According to your comment, we have added “The nomogram may help predict the 1-year, 3-year and 5-year overall survival in patients with primary metastatic adrenal malignancy.” of in the conclusions section of the revised manuscript.

Second, the phenomenon that metastatic neuroblastoma had better oncological survival than metastatic adrenal cortical carcinoma was observed in our study. Therefore, the prognosis of different pathological types is completely different for adrenal malignancy, which may remind us to choose different interventions with curative intent for different pathological types of tumors. According to your comment, we have changed this sentence to “Knowledge of these differences in metastatic patterns and pathological types may contribute to designing targeted pre-treatment assessment of adrenal malignancy and making a personalized curative intervention.” of in the conclusions section of the revised manuscript.

Third, diagnosis and treatment do not equate with equivalent benefit for metastatic adrenal malignancy, and further understanding of outcome of adrenal malignancy, especially metastatic adrenal malignancy, might help make reasonable medical decision and save the unnecessary expend on the advanced tumor. According to your comment, we have added this part in the introduction part due to the limited space in the conclusion part.

Fourth, we believe that pre-treatment evaluation and treatment decision-making is a gradual process, with the deepening of understanding of tumor biological behavior, development of new drugs and inspection methods, and the diversification of treatment methods. Therefore, we hope that subsequent research can help better guide pre-treatment evaluation of cancer and determine interventions with curative intent. According to your comment, we have added “Further efforts are still need to investigate the relationship between more comprehensive factors and adrenal malignancy in the future.” of in the conclusions section of the revised manuscript.

Thank you very much for your great comment.

5. Minor points

1) Authors should submit a figure legend for Figure 1 Figure 1

2) Authors should provide image with higher resolution, since image quality is poor

Response: The reviewer’s comment is well appreciated. We have added the figure legend in Figure1. Meanwhile, we have modified the image and upload the corresponding image. Thank you for your suggestion.

Reviewer #3: 

Dear Editor

In the present study, Zhang.et al performed several experiments aimed to prove the impact of different 24 distant metastases pattern on the survival of patients with adrenal malignancy.

The work is well elaborated, and the focus is very interesting. However, I find some parts of the article need to be improved and I have some revisions to add before acceptance. I recommend acceptance after revision.

Major finds

1. The authors should improve the clinical data added the percentage of metastatic cancer at diagnosis. 

Response: Thank you for your reminding. According to your comment, we have added “A total of 2053 eligible patients with primary adrenal malignancy between 2010 and 2017 were identified from SEER database. Among these patients, the metastatic adrenal malignancy accounted for 47.74 %( 980/2053) at the time of diagnosis, including 493 men (50.3%) and 487 women (49.7%). The metastatic information on bone, brain, liver, and lung metastasis was collected in the SEER database.” in the revised manuscript. Thank you for your suggestion.

2. The authors should clarify which is the mean survival of metastases with bone compare to brain

Response: The reviewer’s comment is appreciated. We have added a new table titled “Frequencies of combination metastasis and 3, 5-y OS” in the revised manuscript. In this table, the median survival time of all metastatic forms was presented. Brain metastasis showed worst survival outcomes in single metastasis. Thank you for your suggestion.

3. The author should improve the comment about the “Kaplan analysis” and highlight which was the best survival with different metastatic sites in the text

Response: The reviewer’s comment is well appreciated. We added the new table which clarified the survival rate and median survival. Meanwhile, we added “The metastatic pattern of metastatic adrenal malignancy was exhibited in Table 3. Theoretically there were 15 metastatic forms, including 4 single metastases and 11 combinations of metastases. However, there were no relevant cases in this study in two types of metastatic forms, which were brain and liver metastases and brain and liver and lung metastases. We found bone metastasis was the most common metastasis in single metastatic patients (24.08%), followed by liver (15.71%), lung (12.04%) and brain (0.71%). As for two sites, the highest frequency was observed in patients with liver and lung metastases at 10.51% (103/980). Lung metastasis presented better early survival rate in single metastasis. Patients with brain and lung metastases had the worse survival rate than other metastatic types in two sites metastases. In addition, median OS cannot be concluded in only bone metastasis and bone and brain metastases in Kaplan-Meier analysis.” in the revised manuscript.

Thank you so much for your comment.

4. In the "Conclusion" section the authors should better explain the impact of their work

Response: The reviewer’s comment is well appreciated. Thank you for your constructive suggestion. We do agree with you that it is very necessary to clearly articulate the significance of metastatic patterns for pre-treatment evaluation and clinical interventions.

First, various prognostic values are associated with different metastatic sites, and knowledge of overall survival is very important for pre-treatment evaluation. According to your comment, we have added “The nomogram may help predict the 1-year, 3-year and 5-year overall survival in patients with primary metastatic adrenal malignancy.” of in the conclusions section of the revised manuscript.

Second, the phenomenon that metastatic neuroblastoma had better oncological survival than metastatic adrenal cortical carcinoma was observed in our study. Therefore, the prognosis of different pathological types is completely different for adrenal malignancy, which may remind us to choose different interventions with curative intent for different pathological types of tumors. According to your comment, we have changed this sentence to “Knowledge of these differences in metastatic patterns and pathological types may contribute to designing targeted pre-treatment assessment of adrenal malignancy and making a personalized curative intervention.” of in the conclusions section of the revised manuscript.

Third, diagnosis and treatment do not equate with equivalent benefit for metastatic adrenal malignancy, and further understanding of outcome of adrenal malignancy, especially metastatic adrenal malignancy, might help make reasonable medical decision and save the unnecessary expend on the advanced tumor. According to your comment, we have added this part in the introduction part due to the limited space in the conclusion part.

Fourth, we believe that pre-treatment evaluation and treatment decision-making is a gradual process, with the deepening of understanding of tumor biological behavior, development of new drugs and inspection methods, and the diversification of treatment methods. Therefore, we hope that subsequent research can help better guide pre-treatment evaluation of cancer and determine interventions with curative intent. According to your comment, we have added “Further efforts are still need to investigate the relationship between more comprehensive factors and adrenal malignancy in the future.” of in the conclusions section of the revised manuscript.

Thank you very much for your great comment.

5. Minor points

The authors should improve the quality of images

The “Ven diagram” needs a color legend

Response: The reviewer’s comment is well appreciated. We have added the color legend in Figure1. Meanwhile, we have modified the image and upload the corresponding image. We are grateful to the constructive suggestions on our manuscript from you.

---

## [Decision Letter · Decision Letter 1]

5 Nov 2021

PONE-D-21-11886R1

The prognosis of different distant metastases pattern in malignant tumors of the adrenal glands: A population-based retrospective study

PLOS ONE

Dear Dr. Zhang

Thank you for submitting your manuscript to PLOS ONE. After careful consideration, we feel that it has merit but does not fully meet PLOS ONE’s publication criteria as it currently stands. Therefore, we invite you to submit a revised version of the manuscript that addresses the points raised during the review process.

We look forward to receiving your revised manuscript.

Kind regards,

Filomena de Nigris, M.D., Ph.D.

Academic Editor

PLOS ONE

Journal Requirements:

2. Did you receive any third party support in conducting this research, analyzing the data, or preparing the manuscript for submission? If yes, provide details as to the organization(s) involved and their specific contributions.

Reviewers' comments:

Reviewer's Responses to Questions

**Comments to the Author**

1. If the authors have adequately addressed your comments raised in a previous round of review and you feel that this manuscript is now acceptable for publication, you may indicate that here to bypass the “Comments to the Author” section, enter your conflict of interest statement in the “Confidential to Editor” section, and submit your "Accept" recommendation.

Reviewer #1: All comments have been addressed

Reviewer #2: All comments have been addressed

Reviewer #3: All comments have been addressed

Reviewer #4: (No Response)

2. Is the manuscript technically sound, and do the data support the conclusions?

Reviewer #1: Yes

Reviewer #2: Yes

Reviewer #3: Yes

Reviewer #4: Yes

3. Has the statistical analysis been performed appropriately and rigorously? 

Reviewer #1: Yes

Reviewer #2: Yes

Reviewer #3: Yes

Reviewer #4: Yes

4. Have the authors made all data underlying the findings in their manuscript fully available?

Reviewer #1: Yes

Reviewer #2: Yes

Reviewer #3: Yes

Reviewer #4: Yes

5. Is the manuscript presented in an intelligible fashion and written in standard English?

Reviewer #1: Yes

Reviewer #2: Yes

Reviewer #3: Yes

Reviewer #4: Yes

6. Review Comments to the Author

Reviewer #1: the authors answered all the questions so for me the work is acceptable.

Reviewer #2: (No Response)

Reviewer #3: (No Response)

Reviewer #4: The authors have greatly improved their manuscript by the recent revisions. My suggestions are very minimal, and are made mostly to increase the rigor and reproducibility of the manuscript.

For all listed p-values (both in the text and in the tables), please provide the hypothesis test method that was used to obtain those values.

For age, it appears that data is reported as median +/- IQR, which is appropriate if the data is not normally distributed or skewed. However, it appears the hypothesis test method used on the age data is a student's t-test which has the assumption that that data is normally distrubted. This is a discrepancy. The authors should check whether the age data is normally distributed. If so, a t-test is appropriate and mean +/- SD should be reported. If the data is not normally distributed, then a Mann Whitney test should be used and median +/- IQR should be reported.

There appear to be some variables that have very small counts (less than 5). It is unclear why a chi-square test was used to analyze these instead of the more appropriate fisher exact test. Please rectify or indicate why the chisquare test is appropriate for all categorical variables, even when the sample size is small.

For Table 2, please indicate what hypothesis test was used to obtain each p-value.

For Table 3, what is the difference between NA & /? Why is median OS unable to be calculated for 2 or more situations?

For the Cox Proportional Hazards models, it would be nice to know why/how the variables were chosen to be included/excluded from the modeling. For instance, there are far more variables reported in Table 1 than used in the Cox modeling. It is unclear why this is so. Please justify.

7. PLOS authors have the option to publish the peer review history of their article (what does this mean?). If published, this will include your full peer review and any attached files.

Reviewer #1: No

Reviewer #2: **Yes: **Concetta Schiano

Reviewer #3: **Yes: **Laura Mosca

Reviewer #4: No

---

## [Author Response · Author response to Decision Letter 1]

17 Dec 2021

We would like to express our sincere gratitude to the reviewers for their constructive and positive comments. Related images can be found in file labeled "Response letter for PLOS ONE''.

Reviewer #4: 

The authors have greatly improved their manuscript by the recent revisions. My suggestions are very minimal, and are made mostly to increase the rigor and reproducibility of the manuscript.

1. For all listed p-values (both in the text and in the tables), please provide the hypothesis test method that was used to obtain those values.

Response: The reviewer’s comment is well appreciated. Thank you for the reminder. 

First, we used chi- Square test in table2. Depending on the conditions of use, the p-values we obtained were derived from Pearson's chi-square test as well as Fisher exact test. Pearson’s chi-square test and Fisher’s exact test are tests of categorical variables. Pearson’s chi-square test assumes that the data: 1) are a simple random sample, 2) are of sufficient sample size and 3) are independent of each other. When cells are sparsely populated, considered any cell value is less than 5 with the traditional rules, Fisher’s exact test was employed [PMID: 28295394]. To be more precise, we have added corresponding annotation of different types of statistical methods in table2.

Second, COX regression model was used in table4 and table5. COX regression models can well address the diversity of survival time distributions in survival data, and can better perform censored data. We have used univariate COX regression analysis in table4 and multivariate COX regression analysis in table5. 

Third, Kaplan-Meier method is often applied to estimate the probability of survival, and it was used in figure2 and figure3. We analyzed the data with log-rank test, which was considered an appropriate test for survival analysis [PMID: 28962743].

Thank you very much for your great comment.

2. For age, it appears that data is reported as median +/- IQR, which is appropriate if the data is not normally distributed or skewed. However, it appears the hypothesis test method used on the age data is a student's t-test which has the assumption that that data is normally distrubted. This is a discrepancy. The authors should check whether the age data is normally distributed. If so, a t-test is appropriate and mean +/- SD should be reported. If the data is not normally distributed, then a Mann Whitney test should be used and median +/- IQR should be reported. 

Response: The reviewer’s comment is well appreciated. Thank you for your reminding. 

First, we have checked that the age data is not normally distributed. So we used Mann Whitney test and reported median. You can find the median +/- IQR in Table1. 

Second, we also checked the age data was not normally distributed in table 2 and added median +/- IQR for more completeness.

Third, we reviewed our manuscript under your instruction and found the mistake in Method section. For more rigorous, we revised the sentence into “Categorical variables were compared using chisquare tests, and continuous variables were analyzed by the Mann Whitney test for non-normal distribution” in the article.

Thank you for your suggestion.

3. There appear to be some variables that have very small counts (less than 5). It is unclear why a chi-square test was used to analyze these instead of the more appropriate fisher exact test. Please rectify or indicate why the chisquare test is appropriate for all categorical variables, even when the sample size is small.

Response: The reviewer’s comment is well valuable. Thank you for your suggestion.

First, as your comment, fisher exact test were more appropriate for use when one or more cells has a count of less than 5. We have conducted fisher exfact act for those variables actually. For table2, we selected appropriate P value for different variables according to the different conditions. To better illustrate the statistical method from which the P values are taken, we have added the corresponding P-values annotations as a and b in table 2. As for annotation a, the P value was selected from Pearson chi-Square.

Second, we believe that we have been misrepresented our statistical method in this paragraph. In SPSS 25, chi-Square included Pearson chi-Square, likelihood ratio and fisher exact test. We were going to describe chi-Square was used to analyze rather than Pearson chi-Square. To be more precise, we have changed the sentence into “Categorical variables were compared using chisquare tests, and continuous variables were analyzed by the Mann Whitney test for non-normal distribution.” in the article.

Third, take variable primary site for example, the P value was selected from Pearson chi-Square when comparing the patients with bone metastasis and without bone metastasis. As for brain metastasis, the P value was selected from fisher exact test, as shown in the figure below.

Thank you for your reminder.

4. For Table 2, please indicate what hypothesis test was used to obtain each p-value.

Response: The reviewer’s comment is well appreciated. Thank you for your reminding. 

In table 2, we aimed to verify that there are differences in patients with metastasis and without metastasis for different variables. We used chi-Square test in table2. Depending on the conditions of use, the p-values we obtained were derived from Pearson's chi-square test as well as Fisher exact test. As shown in QUESTION1. 

Take variable gender for example.

First, establishing test hypotheses and determining test levels.H0: the males and females have the same probability of suffering adrenal malignancy with bone metastasis and without bone metastasis,π1=π2.H1: the males and females have different probability of suffering adrenal malignancy with bone metastasis and without bone metastasis, π1≠π2 α=0.05.

Second, calculate the test statistic and obtain the p-value. In this hypothesis test, every cell is more than 5, Pearson Chi-square is reasonable accurate. And we can acquire the P value < 0.001, as shown the figure below.

Third, At the level of α =0.05, reject H0,accept H1, the distribution of gender among the patients with bone metastasis and without bone metastasis was significantly different. 

Thank you for your suggestion.

5. For Table 3, what is the difference between NA & /? Why is median OS unable to be calculated for 2 or more situations?

Response: The reviewer’s comment is well appreciated. 

First, / was used to indicate that the median OS cannot be calculated due to the absence of related cases. The number of cases of brain and liver metastases and brain and liver and lung metastases were zero. Therefore median OS cannot be calculated

Second, NA was used to present that the median OS unable to display even though the number of related cases were sufficient. The median OS is defined as “the length of time from either the date of diagnosis or the start of treatment for a disease, such as cancer, that half of the patients in a group of patients diagnosed with the disease are still alive” in NCI's Dictionary, as shown in the figure below.

The median OS of those two group cannot be concluded because more than half of the patients still alive at the end of the follow up. Take the bone metastasis for example, we redrawn the survival curve, as shown in the figure below.

In this figure, we added the survival median line, and the curve of bone metastasis are consistent on the median line. So the median OS cannot be calculated in only bone metastasis and bone and brain metastases.

Third，to better present the difference between the NA & /, we have added corresponding annotations below the table. The annotation showed “the median OS cannot be concluded because more than half of the patients still alive at the end of the follow up” for NA and “the median OS cannot be calculated due to the absence of related cases” for /. 

Thank you for your suggestion.

6. For the Cox Proportional Hazards models, it would be nice to know why/how the variables were chosen to be included/excluded from the modeling. For instance, there are far more variables reported in Table 1 than used in the Cox modeling. It is unclear why this is so. Please justify.

Response: Thank you very much for your comment. 

This is a very interesting question. We did not present all variables reported in Table 1 but we have verified and included the variables with statistically significant. The variable such as years of diagnosis, laterality, median income were excluded because the p >0.05 and we commonly assumed that these variables are not significantly different. Other variables such as grade were excluded because the category unknown was unreasonably numerous. So we excluded variables that have no significance in the Cox modeling in order to be more concise and precise.

Thank you for your question.

We are grateful to the constructive suggestions on our manuscript from you.

---

## [Editor Report · Decision Letter 2]

11 Feb 2022

The prognosis of different distant metastases pattern in malignant tumors of the adrenal glands: A population-based retrospective study

PONE-D-21-11886R2

Dear Dr. Zhang

We’re pleased to inform you that your manuscript has been judged scientifically suitable for publication and will be formally accepted for publication once it meets all outstanding technical requirements.

Kind regards,

Filomena de Nigris, M.D., Ph.D.

Academic Editor

PLOS ONE

---

## [Editor Report · Acceptance letter]

4 Mar 2022

PONE-D-21-11886R2 

The prognosis of different distant metastases pattern in malignant tumors of the adrenal glands: A population-based retrospective study 

Dear Dr. Zhang:

I'm pleased to inform you that your manuscript has been deemed suitable for publication in PLOS ONE. Congratulations! Your manuscript is now with our production department. 

Kind regards, 

on behalf of

Prof. Filomena de Nigris 

Academic Editor

PLOS ONE